# Multiple Ways of Nitric Oxide Production in Plants and Its Functional Activity under Abiotic Stress Conditions

**DOI:** 10.3390/ijms241411637

**Published:** 2023-07-19

**Authors:** Chulpan R. Allagulova, Alsu R. Lubyanova, Azamat M. Avalbaev

**Affiliations:** Institute of Biochemistry and Genetics—Subdivision of the Ufa Federal Research Centre of the Russian Academy of Sciences, Ufa 450054, Russia; lubyanova555@mail.ru (A.R.L.); avalbaev@yahoo.com (A.M.A.)

**Keywords:** nitric oxide, NO production, nitrate reductase, molybdoenzymes, non-dedicated NO-forming nitrite reductases, regulation of plant development, stress tolerance, drought, salinity, temperature stress, heavy metals

## Abstract

Nitric oxide (NO) is an endogenous signaling molecule that plays an important role in plant ontogenesis and responses to different stresses. The most widespread abiotic stress factors limiting significantly plant growth and crop yield are drought, salinity, hypo-, hyperthermia, and an excess of heavy metal (HM) ions. Data on the accumulation of endogenous NO under stress factors and on the alleviation of their negative effects under exogenous NO treatments indicate the perspectives of its practical application to improve stress resistance and plant productivity. This requires fundamental knowledge of the NO metabolism and the mechanisms of its biological action in plants. NO generation occurs in plants by two main alternative mechanisms: oxidative or reductive, in spontaneous or enzymatic reactions. NO participates in plant development by controlling the processes of seed germination, vegetative growth, morphogenesis, flower transition, fruit ripening, and senescence. Under stressful conditions, NO contributes to antioxidant protection, osmotic adjustment, normalization of water balance, regulation of cellular ion homeostasis, maintenance of photosynthetic reactions, and growth processes of plants. NO can exert regulative action by inducing posttranslational modifications (PTMs) of proteins changing the activity of different enzymes or transcriptional factors, modulating the expression of huge amounts of genes, including those related to stress tolerance. This review summarizes the current data concerning molecular mechanisms of NO production and its activity in plants during regulation of their life cycle and adaptation to drought, salinity, temperature stress, and HM ions.

## 1. Introduction

Nitric oxide (NO) is an endogenous signaling molecule with free radical nature which performs a wide range of biological functions [1]. Until the early 1990s, NO was considered to be a toxic compound, as significant quantities of it were found in exhaust fumes and industrial waste. A paradigm change concerning the cytotoxicity of NO occurred after the discovery of its role in the regulation of the human cardiovascular system. In this regard, in 1992 NO was recognized as the molecule of the year [2], and in 1998 the American scientists F. Murad, R. F. Farchgott, and L. J. Ignarro were awarded the Nobel Prize. Since that time, a great number of studies have been carried out on the functional activity of NO in various biological systems and it has been undoubtedly proven that it is essential for the normal functioning not only of animals and humans but also all living things, including plants. The ability of NO to perform multiple functions in biological systems is determined by its physiochemical properties. NO is a lipophilic molecule with an unpaired electron on the π-orbital with radical properties and high reactivity, which determines its ability to penetrate easily across cell membranes and rapidly interact with its molecular targets [3,4,5]. When NO interacts with reactive oxygen species (ROS), the generation of reactive nitrogen species (RNS) occurs. Particular attention should be paid to peroxynitrite (ONOO^−^), which is formed by the interaction of NO with superoxide anion radical (O_2_^•−^), and acts as one of the nitrating agents in the transmission of NO signal in plants. In addition, when NO interacts with glutathione, S-nitrosoglutathione (GSNO) is formed, which acts as a storage and transport form of NO capable of transferring it to other biomolecules, in particular to proteins causing their post-translational modifications (PTMs). Several types of PTMs have been identified, among which tyrosine nitration, S-nitrosylation, and metal nitrosylation are considered to be the main ones underlying NO-dependent signaling [6,7,8]. NO-induced PTMs are associated with conformational and therefore functional alterations of target proteins, such as transcriptional factors modulating the activity of various genetic programs, which can explain the extremely wide spectrum of NO regulatory action.

NO participates in the regulation of metabolic processes at all stages of plant ontogenesis from germination and vegetative growth to flowering, fruit ripening, and senescence. NO is involved in the regulation of the cell cycle, differentiation and growth of pollen tubes, morphogenesis, and in the establishment of plant–symbiont interactions [4,7,9,10]. NO deserves special attention due to its participation in plant defense mechanisms in response to biotic and abiotic stress factors, in particular drought, salinity, unfavorable temperatures, and excess of metal ions. This statement is confirmed by the fact that the exogenous application of NO donors contributed to the maintenance of various physiological and biochemical parameters such as seed germination, vegetative growth, photosynthesis, respiration, and water balance in stress-subjected plants [3,11]. Under unfavorable conditions, NO is involved in the regulation of nonspecific and specific protective reactions, inducing the antioxidant and osmoprotective systems of plants, and expression of stress-related genes [7,12]. Despite the fact that, over the past decades, significant progress has been made in understanding the biological activity of NO, the question of the mechanisms of its generation in plants remains one of the most debatable [1,13,14]. The ability of plants to produce NO was revealed in the 1970s, when its emission was detected by gas chromatography from legumes treated with herbicides [7,15]. To date, it has been found that NO production proceeds in plants in a course of multiple redox reactions by arginine-dependent or nitrite-dependent pathways [5,16,17]. The occurrence in plants of multiple ways of NO biosynthesis can be explained by its polyfunctionality and the requirement to maintain NO production at a basic level, ensuring the normal course of different aspects of plant metabolism at all stages of their life cycle in normal conditions or under stress.

## 2. Mechanisms of Nitric Oxide Production in Plants

The whole variety of biochemical reactions, in which the NO molecule is produced, proceeds in plants by two major alternative mechanisms: (1) the oxidative or arginine-dependent and (2) the reductive or nitrite-dependent pathways (Figure 1), and both of them are characterized by a complex nature [5,16,17].

### 2.1. Oxidative Mechanisms

Among the oxidative mechanisms of NO production, the most thoroughly studied are the oxidation of L-arginine and polyamines. Moreover, oxidative generation of NO can occur in plants from hydroxylamine [13,16,18]. The L-arginine-dependent pathway is mediated by the enzymatic activity of nitric oxide synthases (NOSs). During this two-step process, L-arginine interacts initially with molecular oxygen to form N-hydroxy-L-arginine that in turn is converted to citrulline with the release of gaseous NO (Figure 2). The guanidine group of arginine becomes the source of the nitrogen atom of the resulting NO molecule, while the oxygen atom comes from the oxygen molecule participating in the reaction [7,8,12].

The reaction has been well studied in animals, in which it is catalyzed by specific enzymes known as NO-synthases (NOSs, EC 1.14.13.39) [7,12]. Three isoforms of NO-synthases have been identified in mammals: NOS1—neuronal (nNOS); NOS2—inducible or macrophage (iNOS); NOS3—endothelial (eNOS), encoded by three different genes [7,12]. It should be noted that NOS1 and NOS3 are characterized by constitutive expression, while NOS2 is distinguished by inducible activity as indicated by the name of this isoform. Its activation occurs during organism infection and its main functions are connected with protective immune reactions. All isoforms of NO-synthases have a similar structure and operation, being in a homodimeric form. Each monomeric subunit contains two domains: (1) the reductase- or NAD(P)H-FAD-FMN-domain located in the C-terminal region; (2) the oxygenase- or Fe-Heme-domain located in the N-terminal region of the molecule, containing binding sites for heme tetrahydrobiopterin (THB4) and L-arginine. There is a calmodulin (CaM)-binding site between the reductase and oxygenase domains, playing an important role in the regulation of NOS enzymatic activity [7,12,19].

Numerous experimental data in which NO production was analyzed by the accumulation of citrulline, as well as the suppression of NO synthesis in plants by inhibitors of mammals’ NOS-activity, suggest the possibility of NO generation in plants in an arginine-dependent manner [4,8,13,17]. However, NOS-like enzymes similar to mammalian ones have not been identified yet in embryophytes, as evidenced by the large-scale screening of more than a thousand plant transcriptomes [12,20]. Among photosynthetic organisms, NOS-like sequences were found in 15 species of unicellular algae, including *Ostreococcus tauri*, in which the presence of a functionally active NOS enzyme with 45% homology with the NOS sequences of mammals was previously demonstrated [17,19,20,21,22]. Unicellular green algae of the *Ostreococcus* family are the smallest eukaryotic organisms living today on the planet that separated from the common ancestor with higher plants into an independent phylogenetic group in the early stages of evolution. There is an assumption that, in the course of evolutionary development during the transition to a terrestrial lifestyle, green plants have lost genes similar to mammalian NOS genes [12,22]. At the same time, in numerous experimental studies it was clearly demonstrated that the use of inhibitors of mammals’ NOS-activity led to the suppression of NO production and associated biological effects in various species of higher plants [4,13]. Based on these data the existence of certain polypeptides with redox-active domains has been hypothesized, that can be aggregated into a single enzymatic complex, catalyzing reactions of arginine-dependent NO formation in higher plants [4,8,22,23]. Thus, today the task has arisen to identify specific enzymes in higher plants that catalyze the reactions of arginine oxidation leading to production of L-citrulline and the release of NO [14,20,24,25].

Oxidative formation of NO in plants can occur from polyamines (PAs) with the involvement of PAs catabolism enzymes in particular copper amine oxidases (CuAO, EC 1.4.3.22). PAs are low-molecular nitrogen-containing compounds playing a certain role in plant growth and development as well as in stress adaptation [26]. The group of polyamines includes compounds such as spermine and spermidine, which can originate from arginine with the participation of arginase [8,16]. By using two mutant lines of *Arabidopsis thaliana* with over- or down-expression of the arginase gene, it was shown that NO production in these plants depends on the availability of arginine. Arginine-deficient plants were characterized by a reduced level of NO production, which was recovered after treatment with spermine [27]. Exogenous application of spermine or spermidine caused an increased production of NO in the root tips and in the vascular tissues of the leaves in *A. thaliana* seedlings treated with polyamines [16,28]. It is assumed that the generation of NO from PAs can be carried out with the participation of CuAO since the *CuAO1* mutants of *A. thaliana* were characterized by impaired NO synthesis in response to treatment with polyamines [17,22,29]. There are 10 different genes of putative CuAOs that have been identified in *A. thaliana*. It was shown that, along with CuAO1, CuAO8 and polyamine oxidase (PAO, EC 1.5.3.13) can also participate in the oxidation of polyamines to NO [22,30,31].

Hydroxylamine (HA, NH_2_OH) can be another source of NO oxidative production in terrestrial plants. Evidence in favor of this assumption can be found in the data regarding enzymatic NO production from HA in bacterial and animal cells [16,32]. For example, HA-dependent NO formation in nitrifying bacteria may happen with the participation of hydroxylamine reductase [33]. It is also known that NO can be released from HA by interaction with superoxide anion [34]. Moreover, hydroxylamine can be a precursor of L-hydroxyarginine, which is known as an intermediate in the reaction of L-arginine conversion to citrulline with NO release [16,35]. The participation of HA in the production of NO by plants is evidenced by the data obtained on *nia30* mutant lines of tobacco deficient in nitrate reductase with impaired nitrite-dependent NO production. It was shown by gas-phase chemiluminescence that HA addition to the *nia30* suspension culture induced NO production by mutant tobacco cells [36,37]. The physiological significance of NO production from HA is not clear enough, since conclusive evidence of its functioning in plants has not yet been obtained. It is known that NH_2_OH is the first product of ammonium oxidation. In addition, HA formation may be mediated by S-nitrosoglutathione reductase (GSNOR) playing a key role in the maintaining of GSNO homeostasis, which functions as the main storage and transport form of NO in all living organisms including plants [1,16].

### 2.2. Reductive Nitrite-Dependent Mechanisms

The reductive pathways of NO generation are associated with reactions of reduction of nitrate/nitrite that occur or can proceed with the participation of enzymes able to catalyze the one-electron reduction of nitrite to NO. Non-enzymatic nitrite-dependent NO formation was detected in the apoplast of the barley aleurone layer taking place under low pH and high nitrate concentrations [13,38,39]. This reaction was described with the equation:2NO_2_ + 2H^+^ ↔ 2HNO_2_ ↔ NO + NO_2_^−^ + H_2_O ↔ 2NO + ½O_2_ + H_2_O

Non-enzymatic NO production is enhanced in the presence of reducing agents such as ascorbic acid or certain phenolic compounds. It has been suggested that NO produced in the apoplast of barley seeds is involved in the regulation of their germination [39].

Enzymatic reductive NO formation can occur in mitochondria with the participation of components of the mitochondrial electron transport chain (mETC), as well as in chloroplasts, peroxisomes, and in the cytoplasm of plant cells [14,40]. An important role in the reductive process of enzymatic NO production belongs to molybdenum-containing enzymes, including nitrate reductase (NR; EC 1.7.1.1). It is well known that the main functions of NR are associated with the process of nitrogen fixation, in which it catalyzes the reaction of the two-electron reduction of nitrate to nitrite. Moreover, under certain conditions, such as hypoxia and low pH values, it is able to catalyze the one-electron reduction of nitrite to NO [13,16,17].

#### 2.2.1. NO Formation with Participation of Molybdenum Containing Enzymes

The family of plant molybdoenzymes is composed of five members: in addition to nitrate reductase, it includes amidoxime reductase or mitochondrial amidoxime reducing component (mARC), xanthine oxidoreductase (XOR, EC 1.17.3.2), aldehyde oxidase (AO, EC 1.2.3.1), and sulfide oxidase (SO, EC 1.8.3.1). These enzymes perform their own specific functions and at the same time have the potential ability to catalyze nitrite-derived NO formation. Therefore, they are designated in the literature as “non-dedicated” nitrite reductases [41,42].

The nitrite-reducing NO-forming activity of eukaryotic molybdoenzymes is determined by the presence of a specific molybdenum cofactor (Moco), the chemical nature of which was deciphered in 1992 by K. V. Rajagopalan and J. L. Johnson in their pioneering work [43,44,45]. Moco has a unique square-pyramidal structure (Figure 3), formed when the molybdenum atom interacts by thiol bonds with the tricyclic pterin called molybdopterin (MPT) also known in literature as pyranopterin or tetrahydropyranopterin. In general, the square-pyramidal geometry of Moco is stabilized by five coordination bounds between the Mo atom and: (1) one apical oxo-group; (2–3) two sulfur atoms of pyranopterin; (4) one labile –OH/OH_2_ group; (5) one terminal thiol group of cysteine residue or inorganic sulfur atom [41,44,45,46]. Molybdoenzymes have been classified into two families depending on the manner of Moco binding to the protein part of the enzyme molecule. The first is the SO family comprising three members (NR, SO, and mARC); and the second is the XO family including two other members (XOR and AO). In the case of the SO family Moco binds covalently through the terminal thiol group of the Cys residue, or through the terminal sulfur atom in the Moco structure of XO members [41,47].

One of the reaction types catalyzed by molybdoenzymes is the transfer of an oxygen atom from the substrate to water (oxygen atom abstraction) taking place at the expense of the exchange of two electrons at which the oxidation state of molybdenum atom rotates between Mo^6+^ and Mo^4+^ [41,48]. This mechanism underlies the NR-dependent reaction of nitrate reduction to nitrite (Equation (1)).
ONO_2_^−^ + 2e^−^ + 2H^+^ → NO_2_ + H_2_O(1)
NO_2_^−^ + e^−^ + H^+^ → NO + (1/2)H_2_O(2)

A similar process takes place in the reaction of one-electron nitrite-reduction to NO catalyzed by molybdoenzymes in particular by the above-mentioned nitrate reductase (Equation (2)).

##### Role of Nitrate Reductase in the Process of NO Production

Assimilatory NR (Nitrate reductase; EC 1.7.1.1) is a ubiquitous enzyme in plants, fungi, and algae, responsible for the first step of nitrogen fixation and catalyzing the reduction of nitrate to nitrite [13,16,49]. It belongs to NAD(P)H-dependent molybdenum-containing enzymes of the SO family containing the Moco with a characteristic square-pyramidal structure [41]. In plants, the enzyme is found in cytosolic as well as in cytoplasmic membrane-bound forms [24]. As mentioned above the main function of NR is participation in the reaction of two-electron nitrate reduction to nitrite (NO_3_^−^ → NO_2_^−^). Then nitrite is reduced to ammonium with the participation of another key enzyme of nitrogen metabolism, assimilatory nitrite reductase (NiR), which is necessary for subsequent synthesis of amino acids. At the same time, NR involvement in the one-electron reduction of nitrite was found, using NAD(P)H as the electron donor producing the NO molecule (Figure 4). The reaction of NR-dependent nitrite reduction to NO can be presented in an equation:NAD(P)H + 3H_2_O + 2NO_2_^−^ → NAD(P)^+^ + 5H_2_O+ 2NO

It is proposed to designate the nitrite-reducing capability of plant NR leading to NO generation as nitrite: NO-reductase activity (Ni-NR activity) [13,16,50].

At the present time the NR-dependent pathway is considered as one of the main mechanisms of NO production in plants. However, it should be stressed that the role of nitrate reductase in NO generation is not only its direct participation in one-electron nitrite reduction, but also in the production of nitrite itself, which is the main substrate for reductive mechanisms of NO biosynthesis in plants. In addition, NR can mediate electron transfer from NAD(P)H to another molybdoenzyme—mARC, which in turn directly catalyzes the reduction of nitrite to NO, as demonstrated in the model alga *Chlamydomonas reinhardtii* [13,51,52]. Meanwhile, NR is also involved in the utilization of NO and maintaining its homeostasis in plants, since this multifunctional enzyme is able to transfer electrons from NAD(P)H to hemoglobin THB1, which carry out a two-electron conversion of NO to nitrate [22,51].

A three-dimensional configuration of nitrate reductase, which is characterized by a multidomain structure, was proposed in the late 1990s [53]. At the same time new details of the molecular organization and functioning of NR have been revealed in the last two decades [49,54]. The enzyme is able to perform the catalytic activity being in homodimeric form (holo-NR), which is composed of monomers with Mm ~100 kDa and an approximate size of 900 amino acid units. Each monomer subunit harbors three prosthetic groups: (1) Moco-containing domain; (2) cytochrome-b5-containing Fe-heme- (Heme-) domain and (3) FAD-domain composed of FAD-cofactor interacting with the universal cellular reducing agent NAD(P)H (Figure 5). Two flexible regions Hinge1 and Hinge2 are located between the domains. The Hinge1 region which is located between the Moco and Heme domains is a variable amino acid fragment playing an important role in the regulation of NR catalytic activity by phosphorylation of a serine residue. The Hinge 2 region is located between the Heme and FAD domains. In the Moco domain there is a dimerization site that serves to assemble monomers and hold them together to make a homodimer form of enzyme [49,53].

It is known that the potential difference of the redox centers determines the formation of electron transport chains in biological systems. The redox potential values for the three prosthetic groups of the homodimeric form NR (holo-NR) were evaluated using various approaches, in particular by voltammetric methods. It has been established that the redox potential values for the FAD-center is poised from 210 to 190, for the Fe-heme domain from 160 to 130, and for the Moco-center this magnitude is near to 0 mV. Thus, it becomes clear that this system determines the formation of a downward flow of electrons from NAD(P)H to Moco [55]. Furthermore, NR functions as a mini electron transport chain in which electrons are donated from NAD(P)H to FAD and migrated through the Heme domain to Moco where the molybdenum atom is reduced and its oxidation state cycles from +6 to +4. Then the electrons are used in the reaction of nitrate reduction, and in the course of NO generation they can be utilized for nitrite reduction [47,49,52,53].

The participation of NR in nitrite reduction with NO formation was first shown in leguminous plants [56]. Its role in NO production was also revealed later in other species, including wheat [57], *A. thaliana* [58,59,60], tomato [61], and maize [62]. The NO-forming NR capacity is not high and does not exceed 1% of the enzyme’s total activity and depends on various factors such as the ratio of the nitrate and nitrite ions, pH value, oxygen concentration, and post-translational modifications of the enzyme′s macromolecule. First of all, the NR-based NO production determined by nitrite contents can take place at relatively low nitrate and high nitrite concentrations because the affinity of enzyme to nitrite (Km_nitrite_ = 100 mM) is much higher than the normal cell nitrite concentrations (~10 nM∙g^−1^ FW) and exceeds the nitrate inhibition constant (Ki_nitrate_ = 50 mM) [16,63]. Second, NO-producing NR activity increases when the value of cellular pH decreases. It is noteworthy that with a decrease in pH the activity of plastidal nitrite reductase falls with concomitant nitrite accumulation, which promotes NR-inducible NO biosynthesis [16]. Moreover, the post-translational modifications of the enzyme’s molecule impact NR-dependent NO production. Phosphorylation of a conserved serine residue in one of the hinge regions of the enzyme monomer’s subunit causes NR interaction with 14-3-3 proteins leading to subsequent inactivation of the enzyme and its proteolytic degradation [16,54,64].

Different plant species are characterized by the existence of two or more NR isoforms differing by the specificity of functional activity, for example, primary interaction with NADH or NADPH, constitutive or inducible expression pattern. In *A. thaliana* two enzyme isoforms have been identified designated as NR1 and NR2 which are encoded by two different *Nia1* and *Nia2* genes. The fact that NR1 exhibits a higher Ni-NR activity, while NR2 is characterized by a preferential nitrate reductase activity, indicates that various isoforms of the enzyme differ in their specificity of functional activity in plant organisms [22]. Transgenic lines of *A. thaliana* with overexpression of the *Nia1* and *Nia2* genes and increased accumulation of the relevant proteins were distinguished by elevated levels of NO production in comparison with control plants. The significance of NR in the biosynthesis of NO has been proven using *nia1* and *nia2* mutant lines, as well as *nia1nia2* double mutants [49].

It has been shown that NO produced with the NR participation is involved in the regulation of different developmental programs such as root morphogenesis [65], initiation of flowering [66], hormonal sensitivity, and stomatal movements [58,67]. In addition, NR-dependent NO plays an important role in the defensive reactions against pathogens as well as abiotic stress factors such as drought, salinity, HM ions, UV-radiation, and osmotic and temperature stress [49,68,69,70,71].

##### NO Formation by Dual System NR-mARC (NR:NOFNiR)

Until recently, NR was considered as the main enzymatic source of nitrite-dependent NO production in plant organisms [51]. However, the involvement of NR in the generation of NO molecules may be related to the enzyme’s diaphorase activity, which defines by the ability to shuttle electrons from the universal reducing agents NADH or NADPH to its reductase (FAD)- and then cytochrome (Heme)-domains. Then electrons are transferred to another molybdoenzyme, namely mARC, which directly catalyzes the nitrite reduction to NO due to the functioning of the Moco active site in the mARC structure [51,72]. Proteins of the mARC family have been described in prokaryotic and eukaryotic organisms and were first identified due to their ability to catalyze in vitro conversion of certain amidoximes to their active amino forms. Two human mARC isoforms have been identified which is both located on the outer mitochondrial membrane which determines the name of these proteins [46,73,74]. *A. thaliana* also contains two genes for amidoxime reductases *ARC1* and *ARC2*, while the unicellular green algae *Chlamydomonas reinhardtii* has one *ARC* gene and the corresponding protein has cytoplasmic localization [51,75]. ARC proteins of animals and plants are characterized by an identical structure with a molecular weight of approximately 35 kDa performing their activity in monomeric form, unlike other molybdoenzymes functioning only when assembling to homodimers. The mARC structure has one prosthetic group, represented by Moco. Using animal systems, it was shown that the activity of mARC requires the participation of other protein partners, in particular, mitochondrial proteins such as cytochrome b5 reductase (Cyt b5 R) and cytochrome b5 (Cyt b5), and the active center of mARC facing into the cytoplasm. This three-component system is called the Amidoxime Reducing Complex (ARCO) through which electrons are transferred from NADH or NADPH over Cyt b5-R and Cyt b5 to the Moco site of mARC, where the substrate is reduced [46,51,72,74,75].

The substrate specificity of mARC has not been established conclusively. It was found that a wide range of hydroxylamines can be subjected to ARC-dependent reduction [46]. A broad range of hydroxylamines have been identified that can undergo ARC-dependent reduction, among which are the N-hydroxylated forms of nucleotide bases, in particular N-hydroxyaminopurine and N-hydroxycytosine. This fact may indicate the participation of mARC in the neutralization of mutagenic and harmful effects of abnormal bases [51,74]. Another substrate for mARC can be N-ω-hydroxy-L-arginine, which is considered to be an intermediate in NOS-dependent NO generation, and therefore mARC may act as a negative regulator of NOS activity [46].

It has been shown that human proteins hmARC1 and hmARC2 are involved in the production of NO occurring on the outer side of the mitochondrial membrane with the participation of the above-mentioned protein partners of mARC—Cyt b5 R and Cyt b5 [72,76]. It should be noted that the spatial combination of prosthetic groups in the mitochondrial ARCO system has an organization similar to the NR structure and resembles the alignment which can be observed in the intramolecular electron transport chain, in which the reductase (FAD)- and the Cyt b5 (Heme)- domains are also present [46]. Using various NR mutant lines of the alga *Ch. Reinhardtii* with deletions in the Moco or in the reductase and cytochrome domains of nitrate reductase, its role was clearly demonstrated in the electron donations to the Moco-center of ARC which are required for NO synthesis. It was shown that in algal cells both enzymes are localized in the cytoplasm, where NO production was detected. It is important to emphasize that ARC had a high specificity for nitrite in the presence of nitrate concentrations up to 1 mM that is such conditions when NR is unable to catalyze the reduction of nitrite to NO [51]. Taking into the account the critical role of this signaling molecule in the metabolism and vitality of photosynthetic organisms, it was concluded that the main function of plant ARCs is their participation in the process of NO production. For this reason, it was proposed to name plant ARC proteins as NOFNiR (NO Forming Nitrite Reductase), and the two-component NO-producing system formed by NR with ARC was denoted by the abbreviation NR:NOFNiR [13,51,52].

##### Possible Role of XOR, AO, SO in the Process of NO Formation

The ability to reduce nitrite to NO has been identified for molybdoenzymes such as xanthine oxidoreductase (XOR), aldehyde oxidase (AO), and sulfite oxidase (SO), so these enzymes, along with NR and mARC, were included in the group of so-called non-dedicated NO-forming nitrite reductases [41]. Protein molecules XOR and AO have a similar domain organization, and due to the structural similarity of their Moco domains, these enzymes were assigned to the XO family. They are homodimeric flavin-containing proteins with a molecular weight of about 290 kDa. Each monomeric unit contains two iron-sulfur centers [2Fe–2S], one Moco, and one FAD cofactor [42]. The XOR enzyme is ubiquitous in animals and plants and is characterized by a predominantly peroxisomal localization, although in animal tissues it is also detected in the cytoplasm and on the outer membrane of some cells [8,16,41]. It can function in the main enzyme form known as xanthine dehydrogenase (XDH) or in the form of xanthine oxidase (XO) which is formed from XDH, due to reversible oxidation of cysteine residues in its polypeptide chain, such as Cys535 or Cys992 in bovine XOR, or irreversible oxidation of other amino acid residues, such as Lys551 or Lys569 in the same protein [41]. In aerobic conditions both enzyme forms catalyze oxidation of hypoxanthine to xanthine followed by the formation of uric acid. XDH activity induces reduction of NADP to NADPH, while the activity of XO is associated with oxygen molecule reduction followed by superoxide generation, which spontaneously dismutates to hydrogen peroxide [16]. In different independent experiments using such methods as electron paramagnetic spectroscopy, chemiluminescence, or by using NO-selective electrodes, it was shown that XOR isolated from animal tissues is able in vitro to reduce the nitrite with NO formation under anaerobic conditions in the presence of NADH [41,77]. The involvement of XOR in NO production in plant organisms has been demonstrated using allopurinol which is an inhibitor of XOR activity. Its application repressed NO biosynthesis in the roots of white lupine, indicating a high possibility of XOR involvement in the formation of NO in plant organisms [13,78].

Aldehyde oxidases, as well as XORs, have been found in animal and plant cells and characterized by cytoplasmic localization. The specific functions of these enzymes are to catalyze the oxidation reactions of various aromatic and non-aromatic aldehydes with the formation of their corresponding carboxylic acids [74,79]. Four genes of aldehyde oxidases (*AO1*–*AO4*) have been identified in *A. thaliana*. The assembly of their expression products in various combinations leads to the formation of homo- and heterodimers with different substrate specificities. For example, the AO1 homodimer is able to oxidize indolyl-3-acetaldehyde to indolyl-3-acetic acid, which belongs to the hormonal substances of the auxin family. Another AO from *Arabidopsis* known as AOδ, active in an AO2/AO3 heterodimer form, is characterized by high specificity for abscisic aldehyde, which is a precursor of the ABA hormone. The involvement of plant AOs in the biosynthesis of phytohormones may indicate their role in various aspects of plant growth and development, including seed germination, vegetative growth, and adaptation to environmental stress factors [79]. The participation of these enzymes in NO production is evidenced by the data of the experiments in vitro with animal AOs, in which their ability to catalyze the nitrite reduction to NO in the presence of nitrite and such reducing substrates as aldehyde N-methylnicotinamide and NADH [41] has been shown, although AOs’ involvement in NO formation in plant organisms remains to be elucidated. Considering the close structural similarity of AO and XOR, it has been assumed that these molybdoenzymes can play a certain role in plant NO production being the additional enzymatic source of its biosynthesis in photosynthetic organisms [13,16,17].

NO biosynthesis in plants can occur with the involvement of sulfite oxidase (SO) since its nitrite reductase NO-synthesizing capacity has been detected in mammals [41,47,80]. The main functions of SO are associated with the process of sulfite detoxifying when its oxidation to sulfate occurs.
SO_3_^2−^ + H_2_O → SO_4_^2−^ + 2H^+^ + 2e^−^

This reaction is the final step in the oxidative degradation of sulfur-containing amino acids (cysteine and methionine). The SO structure of mammals or birds is characterized by a homodimeric structure and the presence of two prosthetic groups in each subunit: the N-terminal Heme-domain and the C-terminal molybdenum cofactor (unlike NR in which the Moco site is located in the N-terminus). It was revealed that SO in animal cells is localized in the mitochondrial intermembrane space, where the electrons that are released during the oxidation of sulfite are transferred through the Heme-domain to the universal electron acceptor cytochrome c. The SO Heme-domain can be hydrolyzed by partial proteolysis, resulting in a modified form of the enzyme with only one Moco redox-site. This incomplete form of SO is unable to transfer electrons to cytochrome c, but retains the ability to oxidize sulfite, which can be carried out in the presence of artificial electron acceptors [41]. The structure of plant sulfite oxidases was described after identification of the corresponding cDNA in *Arabidopsis* and comparative analysis of plant and animal SO sequences. Herewith the closest similarity reaching about 46% homology at the level of amino acid sequences was found between plant and chicken SO [81]. In addition, their Moco-containing domains were characterized by close structural similarity to that of NR [81,82]. It was found that plant SO also has a homodimeric organization and, like the truncated form of the animal enzyme, does not contain the Heme-domain. It transfers the electrons released during the oxidation of sulfite to molecular oxygen with the formation of superoxide and its subsequent dismutation to hydrogen peroxide. So, it is not surprising that in plant cells SO is localized in the peroxisomal matrix in which the processes of peroxide utilization take place [8,42,74].

It is notably that mammalian SO in the presence of sulfite is able to catalyze the one-electron reduction of nitrite to NO. The SO-catalyzed production of NO occurs at the molybdenum center, and sulfite acts as a reducing agent [41,80]. The reaction rate depends on the medium pH and increases approximately two-fold as the pH decreases from 7.4 to 6.5, although, even at these pH values, the nitrite-reducing activity of SO is significantly lower than that of XO and AO [41]. In addition, SO-catalyzed NO production increases as the oxygen content in the medium drops. Hypoxia constrains the intramolecular electron flow from the Moco of the SO and their movement to cytochrome c, which is accompanied by an increase in electron movement in nitrite and activation of its reduction with NO formation [41,80]. The data on the NO-producing activity of the mammal SO indicate the possibility of this molybdoenzyme involvement in NO generation in plant organisms, which needs to be confirmed experimentally [13,16,47].

#### 2.2.2. Participation of Mitochondrial Electron Transport Chain (mETC) in NO Formation

There is evidence that the reactions of nitrite reduction with NO formation can be associated with the activity of mETC [13,16,83,84]. Mitochondria exhibit nitrite reductase activity and become an important source of NO in the cells of microorganisms, animal and plant tissues under conditions of hypoxia. It has been shown that, in the cells of some fungi, the formation of ATP in the absence of oxygen is based on the use of nitrite as a terminal electron acceptor [16,85,86]. Moreover, ATP production mediated by the reduction of nitrite to NO was revealed in some plant species under anoxia, in particular in the root tissues of barley and rice plants [87]. It was found that mETC enzyme components of complexes III and IV participate in the process of nitrite reduction with NO formation by the means of inhibitory analysis. The use of myxothiazole and cyanide, which are inhibitors of the activity of cytochrome c reductase (complex III) and cytochrome c oxidase (complex IV), respectively, led to the suppression of nitrite-dependent ATP generation associated with production of NO in the roots of barley and rice plants [16,87]. There is an opinion that the process of nitrite reduction to NO in mitochondria during anoxia has a protective effect on their structural organization contributing to maintaining of their activity and, in general, to increasing the viability of plant organisms under unfavorable conditions [13,16,40].

## 3. Nitric Oxide Functional Activity in Plants

### 3.1. Current Approaches to the Study of NO Activity

The study of the functional activity of NO in plants is carried out using pharmacological and genetic approaches with employment of qualitative and quantitative methods for assessing the endogenous NO content [17,88,89,90]. The oldest and best-known quantitative method for estimation of endogenous NO levels is the Griess method [17,91]. A large body of data has been obtained by electron paramagnetic resonance (EPR), gas chromatography, direct and indirect chemiluminescence, and confocal microscopy with the use of diaminofluoresceins (DAFs) and their various modifications, which allow clear detection of the changes in endogenous NO levels and the sites of its formation in plant tissues [17,88,90].

Pharmacological studies are associated with exogenous NO-treatment of plants using NO donors, in particular sodium nitroprusside (SNP), S-nitroso-N-acetylpenicillamine (SNAP), S-nitrosoglutathione (GSNO). Scavengers and/or inhibitors of endogenous NO production are usually used in such experiments to prove a particular effect of NO donors. The best-known scavengers are 2-phenyl-4,4,5,5-tetramethylimidazoline-1-oxyl-3-oxide (PTIO) and 2,4-carboxyphenyl-4,4,5,5-tetramethylimidazoline-1-oxyl-3-oxide (cPTIO). Sodium tungstate (Na_2_WO_4_) and N^G^-nitro-L-arginine methyl ester (L-NAME) are inhibitors of NO production, suppressing the reductive and oxidative pathways, respectively [17,90,92].

Application of NO donors allows one to imitate its effects in plants. At the same time, important data was obtained using mutant and transgenic lines with modified NO production [93,94]. For example, the *nia1* and *nia2* lines as well as the *nia1nia2* double mutants of *Arabidopsis thaliana* are characterized by defects in the *NIA1* and *NIA2* genes encoding two isoforms of nitrate reductase NR1 and NR2, respectively. They were used in investigations of NR involvement in NO production, as well as participation of NR-dependent NO in the processes of root formation, initiation of flowering, and regulation of stomatal movements [49,52,95]. The key role of NR in nitrite-dependent NO production was confirmed in experiments with *NIA1-* and *NIA2*-overexpressing lines of *A. thaliana* characterized by increased NO production [95]. Reduced levels of endogenous NO were detected in *noa1* mutants of *A. thaliana* for the *NOA1* gene (Nitric oxide-associated protein 1), which was initially considered as a plant NO-synthase. Later it was found that it is not a real plant NOS but belongs to the GTPases which has a pleiotropic effect on plant metabolism, in particular on oxidative NO production. That is why *noa1* mutants are widely used in the studies of the regulatory role of NO in plants [94,96,97]. *Nox1* mutant lines with increased NO production with defects in the *CUE1* (chlorophyll a/b binding protein underexpressed 1) gene encoding the chloroplast phosphoenolpyruvate/phosphate translocator have been characterized for *A. thaliana* [98]. Increased arginine content was detected in these mutants, which seems to be responsible for the overproduction of NO in them, but the CUE1-dependent mechanism of NO formation is not yet clear [93]. Nevertheless, *nox1* lines have been successfully used in proteomic experiments for the study of the profile and subsequent identification of NO-modulated proteins [93,98]. At the same time various transgenic lines are used to study the biological activity of NO under normal and stress conditions, for example *A. thaliana* plants with constitutive expression of the rat *nNOS* gene (neuronal NO synthase) and the *OtNOS* gene of the algae *Ostreococcus tauri*, and also *O. sativa* plants expressing the mammalian nNOS gene [93,94,99,100].

### 3.2. The Role of Nitric Oxide in Plant Development

NO has fundamental functions through the whole plant life cycle, being implicated in the regulation of seed dormancy and germination, the cell cycle, vegetative growth, tissue differentiation, root architecture formation, development of symbiotic relationships, flowering, and fruit ripening [7,12,101]. It is characterized by a pronounced seed germination stimulating effect, which has been clearly demonstrated in different plant species, including A. thaliana, barley *Hordeum vulgare* L. [39], lettuce Lactuca sativa [102], wheat *Triticum aestivum* L. [103], etc. Exogenous NO-treatment of Cicer arietinum seeds induced the expression of genes for hexokinase 1, phosphofructokinase 6, pyruvate kinase, α-amylase implicated in the sugar utilization, as well as genes for D4-1-like and B1-4-like cyclins, which are involved in the regulation of the cell cycle [104]. NO donors increased the activity of β-amylase and the level of NADPH oxidation in wheat seeds, inducing hydrolysis of polysaccharides and activation of the pentose phosphate pathway of glucose catabolism, which resulted in stimulation of germination processes [7,105]. An enhancement of β-amylase activity and germination level were also observed in NO-treated seeds of such plant species as soybean *Glycine max* L., barley *Hordeum vulgare* L., maize *Zea mays* L., rice *O. sativa* L., A. thaliana, mustard *Brassica juncea* L., rapeseed *Brassica napus* L. and watermelon *Citrullus vulgaris* [105].

NO is involved in the regulation of vegetative growth. Incubation of root tips of 3-day-old Z. mays seedlings in the presence of such NO donors as SNP, GSNO, sodium nitrite, and nitrocysteine stimulated their elongation [7,106]. NO induced de-etiolation of lettuce L. sativa hypocotyls, reducing their elongation [102]. The presence of SNP at concentrations from 50 to 200 μM in the seed germination medium stimulated an increase in the linear dimensions of 4-day-old wheat seedlings and activation of cell division processes, as evidenced by the mitotic index parameters of the root apical meristem [103]. The important role of NO in the regulation of vegetative growth processes is supported by data on defects in root development, stunted shoot growth, and abnormal flowering in Atnoa1 lines of A. thaliana with reduced NO production [94,96]. NO is involved in the regulation of flower development, which is confirmed, in particular, by data on delayed flowering in A. thaliana plants under the influence of SNP treatment [107]. Moreover, NO-overproducing nox 1 mutants flowered later and NO-deficient nos 1 mutants flowered earlier than the wild type plants [98,107]. NO plays the role of a key component in perception of the photoperiod, as well as in the regulation of pollen tube growth and micropyle orientation [108].

NO is implicated in root growth, acting as a signal intermediate in the auxin-dependent regulation of the development of root architecture [7,101,109,110]. NO treatment of tomato and cucumber seedlings had an auxin-like effect associated with growth suppression in the primary roots and its activation in the lateral and adventitious roots [109,110]. It was shown that SNP-treatment modulated the activity of the cell cycle regulatory genes such as cyclins (CYCA2;1, CYCA3;1, CYCD3;1) and cyclin-dependent protein kinase (CDKA1) in the pericycle cells of tomato which in turn contributed to the induction of lateral root formation [111]. NO induced trichoblast cell differentiation with the formation of root hairs in lettuce plants. An increase in NO content in the root hairs of lettuce plants under treatment with 1-naphthyl acetic acid (NAA) confirms NO participation in auxin-dependent processes of root formation [101,112]. Given that NO plays an important role in the formation of the root system, its involvement in the regulation of mineral nutrition and the establishment of symbiotic interactions is not surprising. NO acts in maintaining homeostasis of macro- and micronutrients, including nitrogen (N), phosphorus (P), potassium (K), magnesium (Mg), zinc (Zn), iron (Fe), etc. [101]. The connection of NO with plant nitrogen nutrition cannot be doubted, since one of the main sources of its production in plants are nitrate/nitrite-dependent pathways in which the key enzyme of nitrogen metabolism assimilatory NR plays the central role [47]. In addition, NO has an effect on nitrate transporter activity, thereby modulating nitrate uptake by the roots [7,101].

NO plays important functions in plant-microbial interactions and in the establishment of symbiotic relationships [7,10]. One of the first pieces of evidence of the involvement of NO in symbiotic interactions is electron paramagnetic resonance (EPR) data on the formation of nitrosyl-leghemoglobin (NO-Lbs) complexes in soybean and chickpea root nodules [113]. NO-leghemoglobin interactions were induced under the treatment of these plants by nitrite solutions [114]. Only young actively metabolizing nodules were able to form NO-Lbs complexes, since they were not detected in senescent structures [113,114,115]. An important role of NO has been revealed in different types of symbiosis, for example, in actinorizobial interaction in the roots of alder Alnus sp., symbiotic interactions in the course of lichen rehydration, and mycorrhizobial symbiosis in olive plants [7,116,117,118].

Summarizing the above evidence, it can be concluded that NO plays an important role in the regulation of the plants’ vital activity at different stages of their ontogeny, confirming the fact that NO is a compulsory member of physiological programs realization under normal growing conditions. At the same time, during the plants’ development optimal conditions interchange with periods of exposure to adverse factors, among which the most widespread are drought, salinity, unfavorable temperatures, and excess of HM ions. Therefore, it is important to consider the role of NO in protecting the plant from these stresses.

### 3.3. The Role of Nitric Oxide in Formation of Plant Stress Tolerance

Numerous studies have shown an enhancement of NO production in plant tissues under different stress factors, such as drought [119,120,121,122], salinity [123,124,125], high or low temperatures [126,127,128,129], HM ions [130,131], UV-radiation, pathogen attack, wounding, etc. [4,7,12,132], leading to a rapid readjustment of the metabolism, contributing to plants’ adaptation to changing environments. Under such conditions, NO functions as a signaling molecule triggering nonspecific reactions as well as specific programs of plant stress tolerance. A common plant response to various stressors is a sharp increase in the production of reactive oxygen species (ROS), which cause numerous types of damage and dysfunction of cellular structures [10]. As the NO molecule possesses free radical properties it can be implicated in the adjustment of cell redox homeostasis by direct interaction with ROS or indirectly through the activation of components of the plant antioxidant system (AOS). The interaction of NO with the superoxide-anion radical (O_2_^•−^) results in the formation of peroxynitrite (ONOO^−^), which belongs to the reactive nitrogen species (RNS) and is characterized by high reactivity. Meanwhile, the level of toxicity of peroxynitrite is significantly lower than that of O_2_^•−^ or H_2_O_2_; therefore, its formation is considered as a direct antioxidant effect of NO [133,134]. It is worth mentioning that peroxynitrite acts as a nitrating agent in NO-induced post-translational modifications (PTMs) of proteins regulating their activity in a changing environment. The indirect protective action of NO occurs by the modulation in the quantitative levels and activities of enzymatic and non-enzymatic components of the plant antioxidant system [12,94]. Along with involvement in antioxidant protection, which can be considered as one of the non-specific defensive mechanisms of NO action, it participates in the maintenance of plant osmotic and water regime balance as well as photosynthetic activity. In addition, NO has an important role in the modulation of the transcriptional activity of numerous genes which are involved in the plant’s tolerance to a wide range of environmental stress factors.

#### 3.3.1. Drought

Drought is one of the most widespread and unpredictable stress factors and has a negative effect on all metabolic processes of plants, leading to inhibition of growth and development and serious losses of crop productivity [90,135]. NO plays a fundamental role in the development of drought tolerance, as evidenced by pharmacological studies with exogenous application of NO donors, as well as information on changes in its endogenous content in plant tissues subjected to drought stress. Increased NO synthesis under water deficit was determined in different plant species such as wheat *T. aestivum* L., parsley *Petroselinum crispum*, pea *Pisum sativum* L. [119,120], sugarcane *Saccharum* spp. [121], *A. thaliana* [122], etc. NO plays a particular role in the regulation of the water regime balance in plants subjected to dehydration, serving as an intermediator in ABA-controlled stomatal movements, which was evidenced by the data on the inability of *Atnoa1* and *nia1nia2* mutants of *A. thaliana* with reduced NO production to show ABA-induced stomata closure under water stress conditions [136]. Moreover, exogenous NO treatment induced stomata closure in the light in epidermal strips of *Vicia faba*. Dark-induced stomata closure in those experiments was significantly decreased by the NO scavenger cPTIO as also by the inhibitor of NOS-like activity L-NAME [137]. NO generation in guard cells in response to drought and ABA production leading to stomata closure was revealed in different species including *Medicago truncatula* [120], *Phaseolus vulgaris,* and *Vigna unguiculata* [138]. By modulating the activity of stomatal movements, NO has an effect on other parameters of the plant water regime. SNP pretreatment improved water use efficiency in *Brassica juncea* plants during PEG-induced dehydration, resulting in maintenance of their growth parameters and higher levels of biomass [139]. Foliar spraying with 25–100 μM SNP contributed to the maintenance of relative water content (RWC) and photosynthetic activity in physalis plants subjected to drought leading to improved growth parameters [140]. Maintenance of the RWC level under NO treatment was also demonstrated in water-stressed *T. aestivum* plants [141].

NO plays an important role in maintaining the redox balance in plants under water deficit conditions, participating in cell protection from oxidative damage [18,90,94,142]. Treatment of *G. max* plants with 100 μM SNP decreased production of peroxide and MDA, activity of lipoxygenase (LOX), and levels of electrolyte leakage under PEG-induced dehydration. Moreover, NO treatment of soybean plants additionally increased the activity of antioxidant enzymes: superoxide dismutase (SOD), catalase (CAT), peroxidase (POD), ascorbate peroxidase (APX), and non-enzymatic antioxidants: flavonoids and tocopherols under dehydration. In alfalfa plants (*Medicago sativa* L.) subjected to water stress, exogenous NO-treatment decreased stress-induced lipid peroxidation, which was accompanied by an enhancement of the gene expression levels of antioxidant enzymes such as glutathione S-transferase (GST), POD, CAT, SOD, and glutathione peroxidase (GPX) [99,143]. Another important aspect of NO protective action is its participation in controlling osmoprotective system activity. Transgenic lines of *O. sativa* with constitutive expression of the rat *nNOS* gene were characterized by increased levels of proline accumulation under exposure to dehydration or salinity [100]. NO treatment increased the proline and glycine-betaine production under drought stress in such plant species as *G. max*, *Lycopersicon esculentum*, *Ginkgo biloba*, and *T. aestivum* [92,141,144,145].

There is evidence of NO participation in the protection of the functional activity of the photosynthetic apparatus. It was shown that NO was involved in the regulation of the expression activity of the *psbA* gene encoding the D1 protein, which is a component of the reaction centers of photosystem II (PS II), which contributed to the normal course of photosynthetic reactions at the grain filling stage in wheat plants under drought [92,141]. SNP treatment increased the expression activity of the *Psa* and *Psb* genes of the PSII and PSII reaction centers, as well as the *Lcha* and *Lchb* genes of the light harvesting complexes, and reduced the levels of chlorophyll degradation under dehydration, which helped to maintain the photosynthetic activity in *M. sativa* L. plants and improve their growth parameters [90,143].

#### 3.3.2. Salinity

Salinity is the next most common stress factor after drought, which is characterized by a double negative effect inducing cell dehydration and intoxication, leading to inhibition of plant growth and productivity. In response to salinity in various plant species, an increase in the levels of endogenous NO was revealed, which is involved in the triggering of adaptive mechanisms to salt stress. Sodium chloride (NaCl) treatment of dune reed callus induced NOS-like activity, as was analyzed by the hemoglobin assay method, and also NO generation which was studied spectrophotometrically by measuring the conversion of oxyhemoglobin to methemoglobin [146]. An increase in NO synthesis under salinity was found in *A. thaliana* [123,147], *Salvinia auriculata* [148], *Helianthus annuus* L. [124], *Olea europaea* L. [149], *B. napus* [125], etc. Using the methods of fluorescence microscopy and fluorometry, an increase in the endogenous content of NO in *S. auriculata* plants was revealed already by 2 h of treatment with 50 mmol L^−1^ NaCl. A positive correlation between NO levels and the content of chlorophylls, carotenoids, and proline was observed in these plants, indicating the involvement of endogenous NO in the activation of salinity tolerance mechanisms in *S. auriculata* [148]. Triple mutants *nia1/2/noa1* of *A. thaliana* were characterized by significantly lower ability to perform NO synthesis and extreme salt-sensitivity, indicating the functioning of arginine- and nitrite-dependent pathways of NO production under salinity and NO involvement in the regulation of plant salt tolerance [150]. At the same time, an increase in endogenous NO content can occur due to a decrease in its degradation. Salinity induced in *A. thaliana* the expression of genes encoding calmodulins AtCaM1 and AtCaM4, which are able to interact with the molecule of S-nitrosoglutathione reductase (GSNOR) and thus reduce its activity, contributing to the increase in NO levels [151,152]. In addition, the participation of NO in plant salt tolerance is evidenced by the data of pharmacological studies. Alleviation of salinity’s negative effects on the various physio-biochemical parameters under exogenous NO application was demonstrated in numerous plant species, in particular wheat [103], rice [153], maize [154], cucumber [155], etc.

NO is involved in the maintaining of ion homeostasis and optimal ratio of K^+^/Na^+^ ions in cells, reducing Na^+^ concentration in the cytosol, facilitating its transport across the tonoplast to the vacuole or across the plasmalemma to the apoplast due to the ability of NO to participate in the regulation of ion channel activity [152,156]. SNP treatment promoted an increase in the K^+^/Na^+^ ratio accompanied by activation of membrane ion pumps such as H^+^-ATPase and H^+^-PPase in root cells of wheat seedlings exposed to NaCl-salinity [157]. NO treatment stimulated the gene expression activity of the Na^+^/H^+^ antiport and contributed to a decrease in the Na^+^/K^+^ ratio in the cells of *Nitraria tangutorum* and *Avicennia marima* plants [158,159]. Treatment of *Brassica napus* cultivars with 100 mM NaCl or 50 mM Na_2_SO_4_ led to an increase in the expression activity of the ion transporter genes *BnCNGC1*, *BnCNGC2* involved in the uptake of Na^+^ ions by cells, the *BnAKT1* gene encoding the inward rectifier K^+^ channel, which under saline conditions exhibits a higher selectivity to the Na^+^ ions in comparison with K^+^, as well as for the sulfate transporter genes *BnSultr4:1* and *BnSultr4:2*. Preliminary foliar spraying with 100 μM SNP of rapeseed plants reduced the salinity-induced levels of *BnCNGC1*, *BnCNGC2*, *BnAKT1*, *BnSultr4:1,* and *BnSultr4:2* genes’ expression and decreased the ratio of Na^+^ and K^+^ ions [125].

NO plays an important role in the regulation of mineral nutrition of plants subjected to saline conditions. SNP-pretreated and salt-stressed *Capsicum annum* plants differed from NO-untreated ones by higher levels of such elements as K, Ca, Mg, Zn, Fe, and B [160]. A positive effect of NO treatment on the content of K^+^, Ca^2+^, and Mg^2+^ was observed in *Gossypium hirsutum* plants when exposed to NaCl-salinity [161]. SNP treatment of rice enhanced in the root cells the salt-inducible transcriptional activity of ammonium transporter genes acting in the NH^4+^ uptake, leading to an increase in nitrogen levels in leaves and seedling biomass accumulation [162].

NO is involved in maintaining the photosynthetic apparatus and the activity of photosynthetic reactions under saline conditions. SNP-treatment of mustard *B. juncea* L. reduced the levels of gas exchange disruption and chlorophyll degradation, and had a protective effect on the thylakoid system stabilizing the structural integrity of chloroplasts under salinity [163,164,165]. SNP-treated *B. napus* plants differed from untreated samples by a significantly lower level of destruction of photosynthetic pigments, including Chl a, Chl b, carotenoids, and anthocyanins under sodium chloride or sodium sulfate salinity [125]. NO contributed to the safekeeping of photosynthetic activity and the quantitative level of photosynthetic proteins: Rubisco large subunit (RBCL) and quinine oxidoreductase-like protein isoform 1 (QOR1) in salt-exposed *A. marina* plants [166]. NO treatment of eggplant *Solanum melongena* L. reduced the inhibitory effect of salinity on the activity of PS II reaction centers [167].

Numerous studies have revealed that NO plays a role in modulating the osmoprotective and antioxidant systems in plants exposed to salt stress. NO treatment of rice, tomato, and mustard plants during salinity led to an additional accumulation of low-molecular compatible osmolytes proline, glycine-betaine, and mono- and oligo-saccharides, which contribute to cell protection from osmotic stress by reducing water potential and normalization of water balance [162,168,169]. Exogenous application of NO donors reduced the levels of lipid peroxidation and ROS production in wheat [103], barley [170], spinach [171], tomato [172], eggplant [167], pea [173], mustard [165], rapeseed [125], and many other plant species. Furthermore, the stimulative effect of NO has been shown on antioxidant enzymes including SOD, APX, glutathione reductase (GR), CAT, and PO, as well as on the activity and quantitative levels of non-enzymatic antioxidants, including proline, glutathione, ascorbate, phenolic compounds, flavonoids, and carotenoids [125,152]. SNAP treatment caused an additional increase in the expression of antioxidant enzyme genes in chickpea plants exposed to salinity [174]. At the same time, NO plays a role in the regulation of the expression activity of many other genes involved in plant defense reactions under salt stress. NO treatment enhanced in *O. sativa* plants subjected to salinity stress the transcription levels of *AMT* and *SUT* genes encoding ammonium and sucrose transporters, respectively, improving young seedlings by supplying them with energy and structural components necessary for their active growth [162]. *O. sativa* plants overexpressing the rat *nNOS* gene with increased NO production were characterized by enhancement in the gene expression of transcription factors OsDREB2A and OsDREB2B, as well as late embryogenesis proteins OsLEA3 and OsRD29A, involved in the plant’s tolerance to salinity and other abiotic stresses [100,152].

#### 3.3.3. Temperature Stress

NO has important signaling functions in the establishment of plant resistance to unfavorable temperatures, both hypo- and hyperthermia [175,176]. An increase in endogenous NO production in response to low-temperature stress was revealed in *P. sativum* [127], *A. thaliana* [128,177], *T. aestivum* L. [178], *Chorispora bungeana* [179], *B. juncea* [129], *Citrus aurantium* [180], and in other plant species. Under low-temperature stress, NO can be synthesized in nitrite-dependent or arginine-dependent mechanisms, but not by these two pathways simultaneously [175]. For example, nitrite-dependent NO formation in *A. thaliana* has been proved using *nia1nia2* double mutants, which differed from wild-type plants by lower levels of NO accumulation and increased sensitivity to low-temperature (4 °C) exposure [128]. Nitrite accumulation and stimulation of enzymatic as well as gene expression activity of NR have been found under low-temperature conditions in *A. thaliana* and *C. aurantium* plants [128,180]. Moreover, mutant lines *Atnoa1/rif1* (for nitric oxide associated1/resistant to inhibition by fosmidomycin1) with inhibition of NOS-like activity did not differ from wild-type in the levels of stress-induced NO production and cold tolerance, additionally indicating nitrite-dependent NO production in *A. thaliana* plants exposed to low temperature [128]. At the same time, treatment of pea leaves and cell cultures of *Ch. bungeana* with an inhibitor of NOS-like activity suppressed NO production during hypothermia, indicating the realization of the oxidative mechanism of NO formation in these plants [127,179]. Overexpression of the *CsNOA1* gene in *Cucumis sativus* plants promoted the accumulation of starch and soluble sugars and a decreased the index of cold damages, while inhibition of *CsNOA1* activity caused the opposite effects. Expression of the cucumber *CsNOA1* gene in *noa1* mutants of *A. thaliana* increased endogenous NO levels in them and helped to the recovery of phenotypic abnormalities caused by NO deficit [181].

NO functioning in the promotion of plant defense responses to low-temperature stress was confirmed in experiments with the exogenous application of NO donors. SNP treatment of *T. aestivum* L. seedlings increased their cold resistance, stimulating activity of the antioxidant system [182]. Activation of the antioxidant system due to the exogenous application of NO was observed in bermudagrass *Cynodon dactylon*, contributing to the reduction in the damaging effects of low temperature on the integrity of cell membrane structures and the operating of the photosynthetic apparatus [183]. In pharmacological experiments using NR inhibitors, NO scavengers, and NO donors it has been shown that NR-dependent NO production was positively correlated with tolerance of *Arabidopsis* seedlings to freezing. Moreover, *nia1nia2* mutants were more sensitive to freezing than wild-type plants. In wild-type plants low-temperature stress up- and down-regulated expression of *P5CS1* and *ProDH* genes, encoding Δ1-pyrroline-5-carboxylate synthetase and proline dehydrogenase, respectively, resulting in enhanced accumulation of proline (Pro). Cold-induced Pro accumulation was reduced by the NR inhibitor and NO scavenger, while the inhibitor of NOS-like activity had no effect on Pro accumulation. Furthermore, cold stress elevated the expression levels of the *ProDH* gene in *nia1nia2* mutants, leading to less accumulation of proline than in wild-type plants, demonstrating that NR-dependent NO production plays an important role in osmotic adjustment during cold acclimation [128].

It has been determined that low-temperature stress increases the level of NO-induced protein PTMs. More than 30 proteins have been identified in *B. juncea* and *A. thaliana* plants that undergo S-nitrosylation, among them those associated with photosynthesis, such as the D1 protein of photosystem II, as well as the large and small subunits of Rubisco [129,184]. It is hypothesized that S-nitrosylation may contribute to stabilization of the D1 protein, acting as a protective mechanism in the maintaining of the structural integrity of PSII. Furthermore, S-nitrosylation can lead to a decrease in Rubisco activity, protecting its protein configuration under conditions of the low-temperature stress [175].

At the same time S-nitrosylation can take place in antioxidant enzyme molecules, as was shown for iron-containing superoxide dismutase (Fe-SOD) of *B. juncea* plants exposed to cold stress. Moreover, exogenous NO treatment increased the activity of Fe-SOD in *B. juncea* plants subjected to low temperatures, which may be due to NO-induced PTMs of the enzyme’s protein molecule [129]. S-nitrosylation also stimulated enzymatic activity of dehydroascorbate reductase (DHAR) and glutathione-S-transferase (GST) under low-temperature stress [185,186].

NO plays a fundamental role in the remodulation of the activity of the genetic programs of plants under the impact of low-temperature stress. The well-known cold-sensitive genetic markers are the genes of the transcription factors which belong to the CBF/DREB (C-repeated Binding Factor/Dehydration Responsive Elements-Binding proteins) family, involved in the regulation of the COR (cold-responsive) genes [187]. CBF genes of *A. thaliana* and *S. lycopersicum* identified as NO-inducible [177,188]. In addition, inhibition of NO production under the influence of a NO scavenger or due to mutations in the NR led to a decrease in the level of cold-induced expression of the *CBF1* and *CBF3* genes in *A. thaliana*. This had a negative effect on the expression of CBF-dependent genes encoding the COR15a, LTI30, and LTI17 proteins, which are involved in the protection of biopolymers and supramolecular cellular structures during low-temperature plant acclimation [175,177,187].

Fine tuning of NO homeostasis is an important factor in the development of plant resistance to high temperature stress [176]. Two mutant lines *hot5 (sensitive to hot temperature 5)/atgsnor1* and *noa1* of *A. thaliana* with an increased level of S-nitrosothiols and excessive NO production, respectively, are characterized by inhibited ability to heat stress adaptation, which can be partially recovered with exogenous treatment by NO scavenger—cPTIO [189]. Mutants of *A. thaliana* with reduced NO production such as *noa1* and *nia1/nia2* are also characterized by extra sensitivity to heat stress, and the thermotolerance of these lines can be enhanced by exogenous treatment with NO donors [190]. There are data in the literature on an increase in NO accumulation in different plant models in response to high-temperature stress [126]. Stimulation of NO synthesis under the influence of hyperthermia was revealed in tobacco leaves, in seedlings of alfalfa *M. sativa* L., and *A. thaliana* [190,191,192]. However, endogenous NO levels decreased during hyperthermia in the protoplasts of tobacco guard cells [193]. High-temperature treatment induced NO generation in the reed *Phragmites communis* Trin., plants of a thermotolerant but not thermosensitive ecotype [194]. The differences in endogenous NO production in response to high-temperature stress revealed in various studies can be explained by the physicochemical features and highly dynamic nature of the NO molecule, as well as by specifics of the experiment and the object of research [126,195].

The positive effect of NO treatment under heat stress conditions has been shown in numerous studies in various plants, such as *O. sativa* [153], *N. tabacum* [192], *A. thaliana* [190], *Z. mays* [196], *T. aestivum* L. [197,198], and *S. lycopersicum* [199]. Preliminary treatment of maize seedlings with a 0.15 µM solution of SNP increased their survival level under the impact of heat stress (48 °C) for 18 h, decreasing stress-induced electrolyte leakage and MDA production [196]. Additional increased production of ascorbate and glutathione as well as activity of monodehydroascorbate reductase and glyoxylase was induced by SNP pretreatment of wheat seedlings subjected to heat treatment (38 °C) for 48 h, which can contribute to improving heat tolerance in NO-treated plants [197]. The SNP-induced increase in SOD, CAT, and POD activity positively correlated with the resistance of wheat coleoptiles to a 10-min-long high-temperature (43 °C) exposure [200]. NO treatment reduced lipid peroxidation and ROS generation in heat-stressed wheat seedlings, increased the activity of enzymatic and non-enzymatic antioxidants, and helped maintain chlorophyll levels [198]. The positive effect of NO on photosynthetic parameters (chlorophyll content and Rubisco activity) was also revealed in overexpressing the *NOA1* gene lines of *O. sativa* [201]. The application of SNP in combination with Ca^2+^ increased the content of such osmolytes as proline and glycine-betaine in heat-stressed *S. lycopersicum* plants [199]. The osmoprotective NO action on plants at unfavorable temperature can be explained by its participation in the regulation of genes involved in the osmolyte metabolism, as evidenced by the data on the additional SNP-induced activation of *P5CR* gene expression in *O. sativa* plants under heat stress conditions [153]. Moreover, these plants were characterized by an NO-induced increase in the transcriptional activity of the gene for the small heat shock protein HSP26 [153]. Therefore, NO can be involved in the development of plant heat tolerance at the genomic level regulating gene expression and the accumulation of heat shock proteins (HSP). The participation of NO in the regulation of the synthesis and accumulation of the HSP70 protein in tomato plants under hypothermia has been revealed [202]. It has been shown that NO, when it interacts with other signaling molecules (H_2_O_2_ and Ca^2+^), is involved in the stimulation of DNA-binding activity of heat shock transcription factors (HSFs), followed by activation of *AtHsp18.2* gene expression and accumulation of the HSP18.2 protein, which positively correlates with thermo-tolerance formation of *A. thaliana* plants [190,203]. Thus, under high-temperature stress conditions NO is involved in the induction of nonspecific defense reactions connected with antioxidant action, osmotic adjustment, protection of the photosynthetic apparatus, maintenance of fluidity, structural integrity, and membrane functions. NO plays a particular role in the regulation of specific genetic programs associated with the activation of HSFs transcription factors with subsequent induction of gene expression and accumulation of HSPs [176].

#### 3.3.4. Heavy Metals

Current literature data clearly point out that NO plays an essential role in mediating plant responses to HM stress [18,204,205]. The levels of endogenous NO in plant tissues change significantly in the presence of HM ions, as was demonstrated, in particular, in cells of *A. thaliana* subjected to an excess of Fe ions [206] and soybean in the presence of Cd^2+^ [207]. Cd-induced elevation of NO production was detected in the roots of wheat seedlings [131], in the roots, leaves, and suspension cell culture of *A. thaliana* [208,209], and in the root tips of barley [130]. A sharp increase in the NO content in the roots of *B. juncea* and *P. sativum* seedlings occurred during the first hours of treatment with Cu, Cd, and Zn ions at concentrations of 100 µM, which further increased during the next 7 days, after which the endogenous NO levels gradually decreased [210]. On the contrary, the levels of endogenous NO decreased in the roots of *Hibiscus muscheutos* plants exposed to Al^3+^ [211], as well as in *M. truncatula* or *O. sativa* plants exposed to Cd^2+^ ions [212,213]. Nevertheless, the results of the vast majority of studies devoted to investigation of NO production in plants subjected to HM-stress indicate an increase in its endogenous content under those harmful conditions. These data serve as an important argument in favor of the signaling role of NO in triggering plant protective mechanisms to HMs which is also confirmed in the experiments with exogenous application of NO donors [204,205]. The presence of 500 μM SNP in the cultivation medium had a protective effect on the growth of barley seedlings under exposure to 200 μM CuCl_2_, as was judged by their linear dimensions and biomass accumulation [214]. Using atomic absorption spectrophotometry, it was founded that SNP treatment inhibited the uptake of Cu^2+^ by *H. vulgare* plants [214], as well as Cd^2+^ and Pb^2+^ by bamboo (*Arundinaria pygmaea*) seedlings, significantly limiting their transport from roots to shoots [215]. Similar results were obtained in the experiments with different plant species and other HMs. In particular, exogenous NO treatment significantly reduced the accumulation of Cr (VI) in *Z. mays* [216], Cd^2+^ in *H. vulgare* [217], Ni^2+^ and Cu^2+^ in *O. sativa* [218,219]. NO treatment decreased Cd-toxicity in *O. sativa* seedlings, reducing its accumulation in the soluble fraction and in the cell walls of the leaves due to enhancement of Cd^2+^ deposition in the root cell walls. It was found that the redistribution of Cd^2+^ in organs and tissues, as well as its immobilization in the roots, can occur through its binding by pectins and hemicelluloses, since their production in the root cell walls increased under NO-treatment, while the relative content of cellulose decreased [220].

There are multiple evidences that NO has an important role in protecting plant cells from oxidative damage induced by HM stress, controlling the levels of ROS generation. Addition of SNP (500 µM) to the growth medium mitigated the harmful effects of CuCl_2_ (200 µM) on growth and photosynthetic efficiency in the shoots of *H. vulgare* L. seedlings, contributing to alleviation of oxidative stress and lipid peroxidation levels [214]. A decrease in NO-treated barley plants of the Cu-induced production of peroxide (H_2_O_2_), superoxide radical (O_2_^•−^), hydroxyl radical (OH^•^), and MDA was accompanied by inhibition of lipoxygenase activity, stimulation of antioxidant enzymes (SOD, CAT, APX, and GPOX), and by stabilization of the glutathione-ascorbate cycle [214]. SNP treatment contributed to a reduction in Cd^2+^- or Pb^2+^-induced ROS production and lipid peroxidation in the culture of *Arundinaria pygmaea*, which was accompanied by the activation of SOD, CAT, APX, and POD, maintaining of the photosynthetic pigments: Chl-a, Chl-b, carotenoids, and by accumulation of osmoprotectants such as glycine-betaine and proline [215]. NO treatment of the *Ch. reinhardtii* cells and *Sorghum vulgare* seedlings subjected to Cu^2+^ stress stimulated expression activity of the *P5CS* gene involved in proline biosynthesis, the antioxidant and osmoprotective properties of which are well known [221,222]. An important contribution to the NO-mediated protective reactions of the plants subjected to excess HMs can be made by phytochelatins, metallothionenins, and membrane transport proteins of the CDF (Cation Diffusion Facilitator) family, also referred to as MTP (Metal Tolerance Protein). SNP treatment of *O. sativa* seedlings exposed to Pb, Hg, Cr, Cu, and Zn ions enhanced the expression of the genes for metallothioneins OsMT-I-1a and OsMT-I-1b, phytochelatin synthases OsPCS1 and OsPCS2, and metal tolerance proteins OsMTP1 and OsMTP5, which play the principal functions in the maintaining of metal ion homeostasis in plants [223,224]. It is known that metallothioneins and phytochelatins act as metal-binding compounds that form chelate complexes with HM ions, reducing their concentration and toxicity in the plant cells [225]. MTPs are membrane proteins involved in the transport of Zn, Cd, Co, Ni, and Mn ions from the cytosol into the vacuole via tonoplast, or from the cytoplasm to the apoplast across the plasmalemma [223,224].

## 4. Conclusions

The study of nitric oxide in plant organisms has been going on for more than four decades and dates back to the identification of nitrosyl-hemoglobin complexes in the nodules of legumes and their ability to emit gaseous NO [15,113]. A huge amount of experimental work has been carried out, and there is no doubt that NO is an essential component of the plant signaling system and is involved in the regulation of cellular metabolism under physiological and stressful conditions (Table 1). A wide range of developmental processes involve NO, ranging from seed germination to vegetative growth, flowering, root morphogenesis, mineral nutrition, symbiosis, etc. Multiple sources of NO production have been identified in plants, among which NR has been studied most thoroughly. However, the canonical NOS enzymes have not yet been identified in terrestrial plants, although convincing evidence of arginine-dependent NO production has been obtained. In this connection, one of the most important tasks is the identification of enzymes involved in such reactions in higher plants. In addition, it is important to continue research to develop strategies of precise detection of NO production sites in plant cells, organs, and tissues at different periods of their ontogenesis and in response to various external stimuli. Investigations using new technologies such as genome-wide expression analysis in combination with proteomic studies with subsequent identification of target post-translationally NO-modified proteins involved in the realization of NO regulatory action are a subject of particular interest. Taking into account the fact that NO does not function alone, it is necessary to consider its activity in the context of a complex network of plant signaling systems in interaction with other bioactive molecules, phytohormones, and growth regulators. Understanding how NO functions in the extremely complex plant metabolism will contribute to the development of methods for its practical application for amelioration of plant growth, stress tolerance, and improvement of crop productivity, which has important socio-economic value.

## Figures and Tables

**Figure 1 ijms-24-11637-f001:**
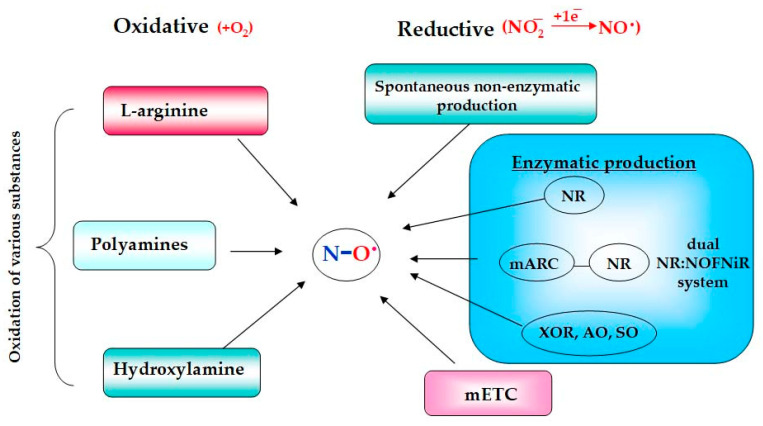
Schematic presentation of the main pathways of NO formation in plants.

**Figure 2 ijms-24-11637-f002:**
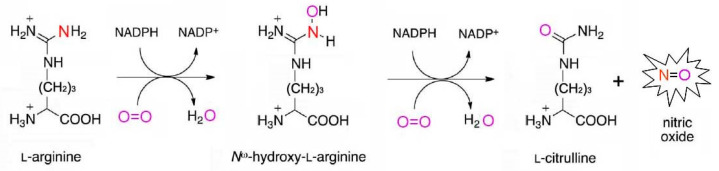
Schematic presentation of a two-step reaction of arginine-dependent NO formation.

**Figure 3 ijms-24-11637-f003:**
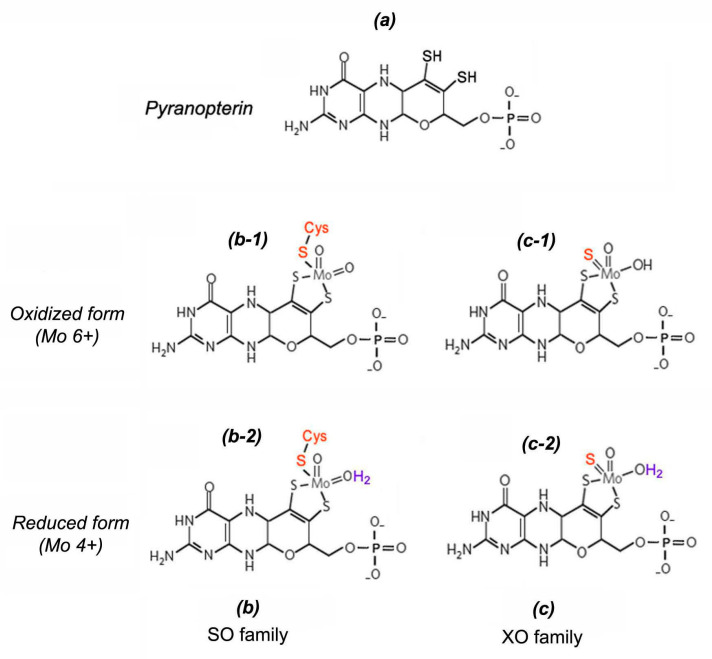
Structural organization of molybdenum cofactor (Moco): (**a**)—structure of pyranopterin-dithiolate heterocycle; (**b**)—Moco structure from molybdoenzymes of sulfite oxidase (SO) family in oxidized form (**b-1**) and in reduced form (**b-2**); (**c**)—Moco structure from molybdoenzymes of xanthine oxidase (XO) family in oxidized form (**c-1**) and in reduced form (**c-2**).

**Figure 4 ijms-24-11637-f004:**
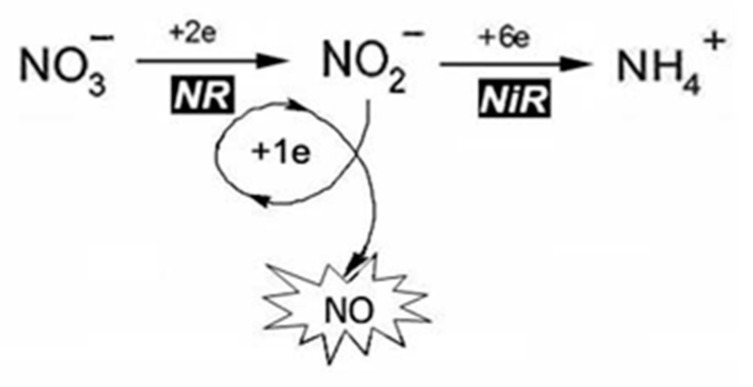
Schematic presentation of NO formation during reactions of nitrogen assimilation.

**Figure 5 ijms-24-11637-f005:**
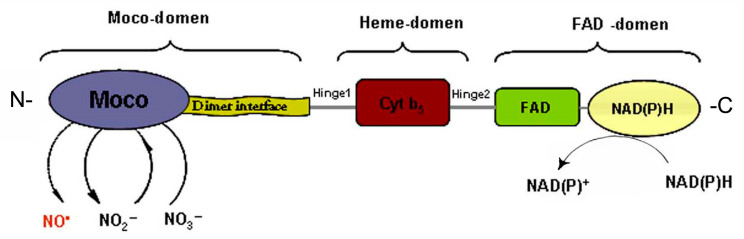
Multi-domain organization of nitrate reductase (NR). Structural FAD-, Heme- and Moco- domains are designated by curly braces at the top of the scheme. Functional sites of the domains are highlighted by different colors.

**Table 1 ijms-24-11637-t001:** Some examples of NO effects in plants.

Plant Species	Modulation of NO Level	Stress Treatment	NO Effects	Reference
*Triticum aestivum, Hordeum vulgare, Glycine max*, *Oryza sativa*, *Zea mays*, *Arabidopsis thaliana*, *Citrullus vulgaris*, *Brassica napus*, *Brassica juncea*	SNP (0.5 × 10^−3^ mol/L); Seed treatment.	−	Increase in β-amylase activity, NADPH oxidation level, activation of polysaccharides hydrolysis and glucose catabolism, stimulation of seed germination.	[105]
*Cicer arietinum*	SNAP (500 µM) SNP (500 µM); Present in the germination medium.	−	Activation of genes for hexokinase 1, phosphofructokinase 6, pyruvate kinase, α-amylase, D4-1-like and B1-4-like cyclins. Stimulation of seed germination.	[104]
*Triticum aestivum*	SNP (200 μM) Present in the germination medium.	−	Stimulation of seed germination, activation of the mitotic index and cell division, increase in cytokinin content, enhancement of shoots and roots growth.	[103]
*Zea mays*	KNO_2_ (10^−7^ M), SNP (10^−10^ M) Present in the germination medium.	−	Regulation of membrane Ca^2+^ channel activity, stimulation of root elongation.	[106]
*A. thaliana*	*Atnoa1* mutants, reduced NO production	−	Defects in root development, stunted shoot growth, abnormal flowering.	[96]
*A. thaliana*	NO-overproducing *nox 1* mutants	−	Delayed flowering.	[107]
	*nos 1* mutants deficient in NO production		Early flowering transition.	[98]
*A. thaliana*	*Atnoa1* and *nia1nia2* mutants, reduced NO production	−	Inability to show stomata closure in response to ABA.	[136]
*Lycopersicon esculentum*, *Cucumis sativus*	SNP (200 μM) Present in the seedlings’ growth medium.	−	Regulation of root morphogenesis (inhibition of primary root growth, activation of lateral and adventitious roots growth), activation of the cell cycle regulatory genes: cyclins (*CYCA2;1, CYCA3;1, CYCD3;1*) and cyclin-dependent protein kinase (*CDKA1*).	[109,110,111]
*Physalis angulata*	SNP (25–100 μM) Foliar spraying	Drought (relative soil water content 20%)	Maintenance of relative water content (RWC), photosynthetic activity, improved their growth parameters.	[140]
*Glycine max*	SNP (100 μM) Foliar spraying	Dehydration, 15% PEG	Activation of antioxidant enzymes (SOD, CAT, POX, APX), accumulation of non-enzymatic antioxidants (phenolic compounds, flavonoids, tocopherols), mitigation of oxidative stress (decrease in MDA, H_2_O_2_, electrolyte exosmosis, LOX activity). Accumulation of osmoprotectants (proline, glycine betaine). Growth improvement.	[144]
*Medicago sativa* L.	Seed germination in the presence of SNP (100 μM)	Dehydration, 10% PEG 6000	Maintaining of growth, water status and chlorophyll levels. Increase in proline accumulation, activation of antioxidant enzymes (SOD, POD, CAT, APX). Modulation of gene expression of transcription factors, photosynthetic proteins, redox homeostasis genes (GST, SOD, GPX, RBOH) and genes involved in phytohormone signaling (ABA, ethylene, auxins).	[143]
*Triticum aestivum*	SNP (0.1 mmol/L)	Salinity NaCl (150 mmol/L)	Regulation of ion homeostasis, increase in the K+/Na+ ratio accompanied by activation of membrane ion pumps H+-ATPase and H+-PPase in root cells.	[157]
*Brassica napus*	100 μM SNP foliar spraying	Salinity NaCl (100 mM) or Na_2_SO_4_ (50 mM)	Reducing the salinity-induced levels of gene expression of ion transporters: *BnCNGC1*, *BnCNGC2*, *BnAKT1*, *BnSultr4:1* and *BnSultr4:2* and decreasing the ratio of Na^+^ and K^+^ ions.	[125]
*Capsicum annum* *Gossypium hirsutum*	SNP treatment	NaCl-induced salinity	Improving the uptake of mineral nutrition elements: K, Ca, Mg, Zn, Fe, B.	[160,161]
*Brassica juncea*	SNP treatment	NaCl-induced salinity	Defense of the photosynthetic apparatus, maintaining chlorophyll levels and protection of chloroplasts thylakoid system.	[163,164,165]
*Triticum aestivum* *Cynodon dactylon*	SNP treatment	Low-temperature stress	Activation of the antioxidant system, protection of cell membrane structures and photosynthetic apparatus.	[182,183]
*Arabidopsis thaliana*.	Inhibition of NO production (*nia1/nia2* mutants, treatment with cPTIO)	Low-temperature stress	Decrease in the level of cold-induced expression of the *CBF1* and *CBF3* transcriptional factors and genes of COR15a, LTI30, and LTI17 proteins.	[175,177]
*Zea mays*	SNP (0.15 µM)	Heat stress (48 °C) for 18 h	Increase in seedlings’ survival level. Decrease in stress-induced electrolyte leakage and MDA production.	[196]
*Triticum aestivum*	SNP pretreatment	Heat stress (38 °C) for 48 h	Increased ascorbate, glutathione production, activation of SOD, CAT, POD, monodehydroascorbate reductase, glyoxylase, maintaining of chlorophyll levels, improvement of growth.	[197]
*Solanum lycopersicum*	SNP treatment	Heat-stress	Increase of the content of osmolytes: proline and glycine-betaine.	[199]
*Oryza sativa*	SNP treatment	Heat-stress	Activation of gene expression of pyrroline-5-carboxylate reductase (*P5CR*).	[153]
*Arabidopsis thaliana*	SNP or SNAP treatment	Heat-stress	Activation of heat shock transcription factors (HSFs), followed by activation of *AtHsp18.2* gene expression and accumulation of the HSP18.2 protein, which positively correlates with plants’ thermo-tolerance.	[190,203]
*Hordeum vulgare*	SNP (500 µM)	CuCl_2_ (200 µM)	Mitigation of the harmful effects of CuCl_2_ (200 µM) on seedlings’ growth and photosynthetic efficiency, decrease in lipid peroxidation levels, alleviation of oxidative stress. Decrease in the stress-induced production of peroxide (H_2_O_2_), superoxide radical (O_2_^•−^), hydroxyl radical (OH^•^) and MDA, inhibition of lipoxygenase activity, stimulation of antioxidant enzymes (SOD, CAT, APX, GPOX) and stabilization of the glutathione-ascorbate cycle.	[214]
*Arundinaria pygmaea*	SNP treatment	Cd^2+^ or Pb^2+^	Activation of SOD, CAT, APX, POD, maintaining of the photosynthetic pigments: Chl-a, Chl-b, carotenoids, accumulation of osmoprotectants: glycine-betaine and proline.	[215]
*Oryza sativa*	SNP treatment	Pb, Hg, Cr, Cu, and Zn ions	Enhancement in the expression of the genes for metallothioneins OsMT-I-1a and OsMT-I-1b, phytochelatin synthases OsPCS1 and OsPCS2, and metal tolerance proteins OsMTP1 and OsMTP5, which all play important functions in maintaining metal ions homeostasis.	[223,224]

## Data Availability

Not applicable.

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
