# Peer review of "Multiple Ways of Nitric Oxide Production in Plants and Its Functional Activity under Abiotic Stress Conditions"

_ijms, 2023, doi:10.3390/ijms241411637_

Round 1

Reviewer 1 Report

This study highlights the role of nitric oxide in plants under abiotic stress conditions and different ways of its production such as molecular mechanisms of nitric oxide formation. The study is well designed and presented. However, some revisions are required to improve the study.

The abstract should be based on brief summary of the work. Like add method of the study and main findings of the study. Current abstract looks like explanation of the topic.

Also highlight which abiotic stress was considered in this study.

Also add conclusion and future perspective in the abstract.

In introduction discuss the role of NO under abiotic stress. 

Paragraph 1st of the introduction some place the author wrote full form of nitric oxide while some used as NO. please be consistent.

Line 148-151 which numerous studies? Lack reference. Also check other places lacking references could be cited with the suggested studies. doi: 10.1016/j.cclet.2021.06.024, doi: 10.1016/j.cclet.2021.09.009

Line 153-155 lack reference and should be cited with the following relevant study https://doi.org/10.3390/antiox12020268

Section 2.2 would be better to add a scheme or illustration of NO production during stress

Section 2.1.1 line 176 should be cited with relevant study. The following study should be cited https://doi.org/10.3390/ijms22179175

Line 182 replace Arabidopsis thaliana with A. thaliana. Also check other places.

The study should include some more information which could be helpful for the readers such as reliable and sensitive methods for the detection and quantification of nitric oxide in plants for example, fluorescent probes, spectrophotometric assays, or imaging techniques specifically designed for nitric oxide detection.

In every stress sections should include enzymatic and non-enzymatic sources of nitric oxide, such as nitric oxide synthase-like enzymes, nitrate reductase, or the nitrate-nitrite-nitric oxide pathway

Use of abbreviations must be consistent. also check typos and grammatical mistakes

Author Response

We are grateful to the referee for the detailed analysis and comments on the review. We tried to take them into account and made corrections to the manuscript.

The abstract has been rewritten; it is based on the brief summary of the work and highlights the abiotic stresses that are discussed in the article.

The full name "nitric oxide" is used at the first mention, which in the following text is denoted as NO.

The necessary relevant references have been added to the content of the article.

The new version of the manuscript cites the following recommended studies: doi:10.1016/j.cclet.2021.06.024; doi.org/10.3390/antiox12020268; doi.org/10.3390/ijms22179175.

The Latin name of plant species at the first mention is written in the full version (Arabidopsis thaliana; Hordeum vulgare L.; Triticum aestivum L.), then the abbreviation is given (A. thaliana, etc.).

The new version of the article contains a section describing methods for NO studying in plants (Section 3.1.).

We included the Table summarizing the main effects of NO on plants.

It is also necessary to note that the structure of the manuscript has been changed radically and now starts with NO biosynthesis and then goes to the NO effect under abiotic stress. This change was made in according to recommendations of Reviewer 2.

Reviewer 2 Report

The review is a huge recopilation on the existing literature on NO in plants. It is worth publishing as it  is very complete and comprehensive report, but it requires major improvements.

1) There is a lot of text and few figures nor tables. The reading is quite difficult as there is a lot of text. I recommend including a table or a figure summarizing the main effects of No under different stresses to help the reader.

2) The order is not logical. I would change the organization of the review and start with No biosynthesis and then go to the NO effect under abiotic stress. Also in the present form, the two parts of the review are very independent. Which biosynthesis pathways are upregulated by abiotic stress? Please include a figure on the regulation of the biosynthetic pathways by abiotic stress.

3) Conclussion should be concise, but is too long and includes a lot of introductory information. Please, shorten the conclusion to a single paragraph comprising the main points of the review and the future prospects for NO biology.     

English is very poor at some points of the paper. For instance in the abstract:

"The review is summarized the data.." instead of "This review summarizes the data..."

Author Response

We are grateful to the referee for the detailed analysis and comments on the review. We tried to take them into account and made corrections to the manuscript.

We included the Table summarizing the main effects of NO on plants.

The structure of the manuscript has been changed and now starts with NO biosynthesis and then goes to the NO effect under abiotic stress. The text has been checked for errors.

In accordance with the recommendations the conclusion has been rewritten completely.

In addition, the abstract and introduction were also changed.

Round 2

Reviewer 2 Report

Paper is now ready for acceptance.